# Workplace Cognitive Failure among Nurses during the COVID-19 Pandemic

**DOI:** 10.3390/ijerph181910394

**Published:** 2021-10-02

**Authors:** Judith E. Arnetz, Eamonn Arble, Sukhesh Sudan, Bengt B. Arnetz

**Affiliations:** 1Department of Family Medicine, College of Human Medicine, Michigan State University, Grand Rapids, MI 49503, USA; dr.sukhesh87@gmail.com (S.S.); arnetzbe@msu.edu (B.B.A.); 2Department of Psychology, Eastern Michigan University, Ypsilanti, MI 48197, USA; earble2@emich.edu

**Keywords:** nurses, cognitive failure, work performance, COVID-19, secondary trauma

## Abstract

Numerous studies provide evidence of the physical and emotional strain experienced by nurses during the COVID-19 pandemic. However, little is known regarding the impact of this occupational strain on nurses’ cognitive function at work. The aim of this study was to identify factors associated with workplace cognitive failure in a sample of U.S. nurses during the COVID-19 pandemic. An online questionnaire was administered in May 2020 to Michigan nurses statewide via three nursing organizations (n = 695 respondents). Path analysis was conducted to test the parallel effects of frequency of contact with COVID patients and personal protective equipment (PPE) supply on workplace cognitive failure scores. Mediation effects of stress, sleep quality, secondary trauma, and work-related exhaustion were examined for each exposure. Results revealed significant indirect effects of all mediators except sleep quality of contact with COVID patients (cumulative indirect effect = 1.30, z = 6.33, *p* < 0.001) and PPE (cumulative indirect effect = −2.10, z = −5.22, *p* < 0.001) on cognitive failure. However, 58% of the PPE effect was direct. To reduce the risk of cognitive failure, healthcare organizations need to provide nurses with protective equipment and work environments that allow nurses to strengthen their resilience to extreme working conditions.

## 1. Introduction

More than one and a half years after the onset of the corona virus disease (COVID-19) pandemic, there is ample evidence of the toll that caring for COVID patients has taken on nurses’ health [1,2]. Multiple studies have reported significant mental health symptoms among nurses involved in direct care of COVID patients around the globe, including China [3,4,5], Iran [6], Israel [7], Italy [8], the Philippines [9], Portugal [10], Spain [11], and the U.S. [12]. Frontline nurses have exhibited high rates of psychological distress [9], depression [3,4,12], anxiety [4,12] and post-traumatic stress disorder [4,7,12]. Mental health problems were shown to be significantly higher among nurses who were not provided with adequate personal protective equipment (PPE) [10,12]. The constant use of PPE during the current pandemic has also been associated with impairments in communication and situational awareness, negatively affecting perception and cognition [13]. At the same time, nurses’ prolonged use of PPE has been associated with physical health problems, including headache [14], respiratory symptoms, and ocular, nasal, and skin problems [15]. Critical care and emergency nurses reported increased workloads and patient-nurse ratios that made it difficult or impossible for nurses to take breaks. Adding to their physical exhaustion was the emotional distress of being unable to provide psychosocial comfort to COVID patients and their family members [11]. Frontline nurses have also reported severe insomnia [4,8,16]. These studies provide substantial evidence of the physical and emotional strain experienced by nurses treating COVID patients. Sustained stress has been linked to neurocognitive dysfunction [17,18]. However, little is known regarding the impact of this occupational strain on nurses’ cognitive function at work. In view of the prolonged duration of the COVID-19 pandemic, a closer look at its implications for nurses’ workplace cognitive function is warranted.

### 1.1. Work Stress and Cognitive Failure

The concept of “cognitive failure” was introduced by Broadbent and colleagues in the 1980’s to represent “lapses” of perception, memory, or action [19]. Cognitive failure research gradually developed from a focus on individual traits (e.g., neuroticism) to an examination of cognitive processes and failures related to one’s work [19,20]. Building on the work of Broadbent and colleagues (1982), Wallace and Chen (2005) developed a questionnaire aimed at characterizing the nature of cognitive failure in the workplace [21]. In subsequent research, cognitive failure was associated with accident proneness in sailors [22]. In studies of nurses, work characteristics in the form of task-related stressors [23] and workflow interruptions [24] were associated with increased workplace cognitive failure. Work stress and cognitive failure have also been associated with nurses’ self-reports of adverse patient events [25,26].

### 1.2. The COVID-19 Pandemic and Cognitive Failure

Nurses as an occupational group were experiencing high levels of work stress, burnout [27,28] and compassion fatigue [29,30,31] well before the onset of the COVID-19 pandemic. During the pandemic, these have increased, especially compassion fatigue [32]. Compassion fatigue is a component of the Professional Quality of Life (ProQuol) measure and encompasses the experience of secondary or vicarious trauma of the people one has helped [33], such as the patients one has cared for. During the COVID pandemic, the combination of intense heavy workloads [11,30], experiencing frequent patient deaths [7], fear of contracting the illness oneself or spreading it among one’s family members [6,34], feelings of physical and emotional fatigue [30], and vicarious trauma [29] may have created a potential breeding ground for cognitive failure. Nurses may be so overwhelmed by the physically and emotionally demanding tasks of caring for large volumes of COVID patients that it may impair their ability to fully focus on the patient care tasks at hand.

### 1.3. Conservation of Resources and Study Hypotheses

The current study aimed to identify factors associated with workplace cognitive failure in a sample of Michigan nurses. The Conservation of Resources (COR) model [35] was used as a theoretical framework for understanding the relationships between individual and work-related factors and workplace cognitive failure. The COR model purports that individuals strive to obtain, retain, and enhance personal resources [35,36] in an effort to thrive. Loss or lack of resources can make it difficult for an individual to interact with their environment and results in stress reactions [35,36]. Hobfoll [35] described “loss spirals” (p. 519) which result when an individual utilizes more resources than can be replenished, leaving them at greater loss in the future [37]. Wallace & Chen (2005) found that work overload, i.e., having too little time to conduct work-related tasks, was related to cognitive failure [21]. It has been suggested that overload of work tasks and/or task complexity can lead to a depletion of an individual’s resources resulting in cognitive failure [36]. With COR theory as a foundation, we deemed it possible that nurses working on the frontlines of COVID care could be experiencing stress and strain, both physical and emotional, and that these reactions could be indicative of a loss of resources. In keeping with Wallace & Chen (2005) [21], nurses’ resource loss could be associated with workplace cognitive failure [19].

Based on the existing literature, our team’s previous research [12,32], and with the COR [35] as a theoretical framework, we hypothesized that nurses working more frequently with COVID patients would experience higher levels of workplace cognitive failure. Assuming that physical exposure to COVID patients would be related to cognitive failure, we also hypothesized that inadequate PPE—yet another physical stressor—would relate to higher cognitive failure. Lack of PPE has been strongly associated with poor mental health [10,12] and stress among nurses [34] in previous studies. Moreover, the effects of these physical exposures on cognitive failure could potentially be mediated by a reduction in nurse resilience via decreased well-being (perceived stress, sleep quality) and increased emotional (secondary trauma) and physical (work-related exhaustion) strain. Higher levels of stress, secondary trauma, and work-related exhaustion and poorer sleep quality could contribute to a loss of resources among frontline nurses, resulting in workplace cognitive failure.

We hypothesized the following regarding contact with COVID-19 patients:

**Hypothesis** **1** **(H1).**
*Frequent contact with COVID-19 patients would be positively related to workplace cognitive failure scores (WCFS).*


Further, we hypothesized that:

**Hypothesis** **1a** **(H1a).**
*The positive relationship between contact with COVID-19 patients and WCFS would be mediated by stress.*


**Hypothesis** **1b** **(H1b).**
*The positive relationship between contact with COVID-19 patients and WCFS would be mediated by sleep quality.*


**Hypothesis** **1c** **(H1c).**
*The positive relationship between contact with COVID-19 patients and WCFS would be mediated by secondary trauma.*


**Hypothesis** **1d** **(H1d).**
*The positive relationship between contact with COVID-19 patients and WCFS would be mediated by work-related exhaustion.*


With regard to availability of PPE, we hypothesized the following:

**Hypothesis** **2** **(H2).**
*Availability of PPE would be inversely related to workplace cognitive failure scores.*


**Hypothesis** **2a** **(H2a).**
*The inverse relationship between availability of PPE and WCFS would be mediated by stress.*


**Hypothesis** **2b** **(H2b).**
*The inverse relationship between availability of PPE and WCFS would be mediated bysleep quality.*


**Hypothesis** **2c** **(H2c).**
*The inverse relationship between availability of PPE and WCFS would be mediated by secondary trauma.*


**Hypothesis** **2d** **(H2d).**
*The inverse relationship between availability of PPE and WCFS would be mediated by work-related exhaustion.*


The conceptual model for these hypotheses is depicted in Figure 1.

Independent of the mediation analyses, the current study also aimed to describe and compare nurses’ experiences of cognitive failure and psychosocial variables in different contexts. We expected that nurses would be caring for greater numbers of COVID patients on emergency and critical care units, and therefore anticipated significantly higher ratings of cognitive failure, stress, secondary trauma, and work-related exhaustion, and significantly lower sleep quality among nurses working in those practice settings.

## 2. Materials and Methods

### 2.1. Study Design

A cross-sectional online survey was conducted in a sample of Michigan nurses in May 2020. The study was determined exempt by the Institutional Review Board at Michigan State University (MSU Study ID: STUDY00004459). 

### 2.2. Participants

Participants were recruited from the American Nurses Association (ANA) Michigan, the Michigan Organization of Nurse Leaders (MONL), and the Coalition of Michigan Organizations of Nursing (COMON). All members of the three organizations (approximately 18,300) and their colleagues were eligible to participate. ANA Michigan distributed surveys directly to nurse members while COMON and MONL used snowball recruitment, asking members to distribute the survey within their respective organizations. The three organizations sent an emailed invitation including a link to the Qualtrics survey. Nurses were informed that the survey was anonymous and confidential; those who agreed to participate completed a consent statement in Qualtrics before responding to the survey [12,34]. 

### 2.3. Study Variables

The 85-item questionnaire included questions on individual and work-related demographics, COVID-19 experience, workplace emergency preparedness, mental health and well-being, and work-related stress, exhaustion, secondary trauma, and cognitive failure [12,34]. 

#### 2.3.1. Independent Variables

Individual and work demographic variables included age, gender, race, having a management position, and practice setting. Work-related COVID factors included frequency of contact with COVID-19 patients, assessed by a single item on a four-point scale (Never to Very often) and access to adequate PPE, also measured by a single item measured on a four-point scale (Not at all to Definitely; not applicable could also be selected). 

Well-being factors included stress and sleep quality, and were measured by responses to validated single-item 0–10 visual analogue scales (VAS), “How do you rate your current stress level?” and “How do you rate your current sleep quality? These and similar single-item VAS items have been shown to have excellent construct [38] and predictive validity [39]. 

Emotional factors included an item measuring nurses’ experience of secondary trauma, measured by a single item from the Compassion Fatigue subscale of the ProQuol [33], “I feel as though I am experiencing the trauma of the patients I care for.” It was measured on a five-point scale from Never (1) to Very often (5). Work-related exhaustion was assessed by a 3-item subscale from the Quality Work Competence (QWC) questionnaire [40] on a five-point scale ranging from Never (1) to Very often (5). The items are, “I feel emotionally drained after work,” “I feel worn out after work,” and “I feel tired when I think about work.” Responses were summed to a total score with higher values indicating greater work-related exhaustion [40,41]. The QWC questionnaire has been used in a large number of studies (e.g., [41,42]), has excellent reliability and validity and has been validated against biological markers of stress and inflammation [43]. 

#### 2.3.2. Dependent Variables

The main study outcome was workplace cognitive failure, assessed using the Workplace Cognitive Failure Scale (WCFS) [21]. The WCFS is a 15-item scale assessing the frequency of experiencing lapses in memory, attention or physical action during work ranging from Never (1) to Very often (5). A sample item is “Cannot remember whether you have or have not turned off work equipment.” Only the composite score was used in the current study. Scores for the overall scale were calculated by summing the component items, with higher values indicating more symptoms of cognitive failure. The Cronbach’s alpha for the total scale in this study was 0.91. 

### 2.4. Data Analysis

Statistical analysis was conducted using IBM SPSS statistics, V.27, 2020 (IBM Corp, Armonk, NY). A two-sided *p* value < 0.05 was deemed to represent statistical significance. Chi-square tests were used to determine whether there were significant differences in demographics (age, gender), pandemic-related factors (contact with COVID-19 patients, provision of adequate PPE), and emotional factors (secondary trauma) by practice setting. The analyses excluded “not applicable” responses to the PPE question (n = 55). Bivariate analyses using Pearson’s r were used to measure correlations between the continuous variables stress, work-related exhaustion, sleep quality, and WCFS. Analysis of covariance, controlling for age, with planned comparisons and Bonferroni correction was used to test for significant differences in WCFS, sleep quality, stress, secondary trauma, and work-related exhaustion by practice setting.

Mediation hypotheses were tested in a path analysis specified with MPlus (v. 7.4) (Muthen & Muthen, Los Angeles, CA, USA) [44]. Observed and summary scale variables of each construct were included to test the direct effects of contact with COVID patients and PPE on WCFS, and their mediated effects via stress, sleep, secondary trauma, and work-related exhaustion. The parallel, multiple mediation model was specified to test Contact and PPE as parallel effects, mutually mediated by a set of secondary emotional and physiological responses in the workplace. Age and gender were tested as covariates of all effects. The model was fit to the data with maximum likelihood estimation with robust standard errors, which does not require multivariate normal distributions for valid estimates [45,46], and as applied here is appropriate for a model that includes variables measured on ordinal and continuous scales [44]. The model was fit to the data from the entire sample (N = 692), including the 153 participants (22%) who did not respond to at least one survey item of the analyzed constructs. Data were missing not at random (Little’s χ^2^ (20) = 75.62, *p* < 0.001); individuals who completed all items were more likely to be younger than those who did not (χ^2^ = 5.23, *p* = 0.02) but were similar in gender representation (χ^2^ = 0.26 *p* = 0.61). Maximum likelihood estimation with robust standard errors provides unbiased estimations with ordinal scales [47] without the need for data imputation and maximizes external validity of the hypothesis test [48]. Maximum likelihood estimates are robust under the assumption of data missing at random, and by including age as a covariate in the model the pattern of missing data is taken into account and the assumption is met.

The quality of the model was evaluated by a compendium of accepted fit indices [49]: chi-square non-significance, Comparative fit index (CFI) greater than 0.9, and standardized root mean residual (SRMR) and root mean squared error of approximation (RMSEA) each less than 0.05 indicate for excellent model fit. The quality of the regression procedures was further evaluated for possible outliers, confirming multivariate normality met when Cook’s distance is less than 3 for all cases [50].

All path coefficients and correlations are reported with unstandardized estimates that can be interpreted by the original scale, and standardized estimates that can be compared for differences in effect magnitude between paths. Mediation was tested by indirect effect estimation [51], including significance testing with a Sobel z-test (α = 0.05) and 95% confidence intervals of unstandardized effects, which if not overlapping zero further supports the mediated effect [52].

#### Power Analysis

The study includes data from a convenience sample. Sensitivity analysis with simulations in G*Power (Faul, Erdfelder, Lang & Buchner, Dusseldorf, Germany) [53] indicate the sample size is sufficient to provide 80-95% power to detect at least small omnibus group differences in analysis of covariance (ANCOVA, α = 0.05, η_p_^2^ = 0.02) and moderate effects in planned comparisons with Bonferroni correction (α’ = 0.01; d = 0.41–0.69) to significance. In addition, the sample size provided sufficient sensitivity to test small effects in linear regression for mediation analyses: f^2^ = 0.02–0.03, power = 80–95%, α = 0.05. 

## 3. Results

A total of 695 nurses responded to the survey. An exact response rate could not be calculated due to the use of the snowballing recruitment technique, allowing nurse members to reach out to non-member colleagues. Nevertheless, based on the total membership of 18,300 nurses in ANA-Michigan, MONL, and COMON, our response rate was approximately 4%. No significant differences were found between the respondent sample and the total population of Michigan nurses (MPHI 2019) in terms of gender or ethnicity [12]. Missing values for the study variables ranged from 1% to 16% and were highest for the WCFS items (16%, n = 111). Approximately one third of nurses who did not respond to these items (n = 30, 27%) were retired or unemployed. A comparison of nurses with missing (non-response) and complete responses on the WCFS revealed that non-respondents had significantly less contact with COVID patients (*p* < 0.001). Some degree of cognitive failure (sometimes, often, or very often) was endorsed by 76.9% (n = 453) of nurses. Endorsement of individual cognitive failure items (sometimes, often, or very often) ranged from a high of 47% (n = 276) for being easily distracted by coworkers to a low of 1.3% (n = 8) for accidentally starting or stopping the wrong machine.

Characteristics of the study participants are shown in Table 1. Most nurses were female (93.6%, n = 644), Caucasian (87.9%, n = 611) and more than half (55%, n = 376) were 45 years or older. More than one third of nurses worked in inpatient acute care (32.8%, n = 220), 27.6% (n = 185) worked in primary/ambulatory care, 15.2% (n = 102) worked in emergency or critical care, and 8.9% (n = 60) worked in pediatrics or obstetrics. The remainder (15.5%, n = 104) worked in other practice settings, which included hospice/home care/long-term care (n = 32; 4.8%), mental health (n = 18; 2.7%) or academic (n = 54; 8%). Fifteen nurses (2.2%) were unemployed and 15 (2.2%) were retired. Approximately 40% (n = 269) reported frequent contact with COVID-19 patients and 24.9% (n = 163) reported that their workplace did not provide adequate PPE. About 34.6% (n = 214) reported experiencing secondary trauma from the patients they cared for sometimes, often, or very often.

Table 2 depicts results from chi-square tests examining differences in discrete study variables by practice setting. Nurses employed in emergency/intensive care (ICU) units tended to be younger (<45, 65.3%) while nurses employed in pediatrics/obstetrics (63.3%) or primary care (66.8%) settings were older (≥45 years, *p* < 0.001). Those working in emergency/ICU had significantly more frequent contact with COVID-19 patients (81% often and very often, *p* < 0.001) and reported experiencing secondary trauma from the patients they cared for often/very often (14.6%, *p* < 0.001) compared to other settings. There were no significant differences across settings in gender or workplace provision of PPE.

Descriptive statistics and bivariate correlations between continuous study variables and cognitive failure symptom scores are shown in Table 3. The mean score for the total WCFS was 28.75 (SD 8.66). Mean scores for stress were 5.95 (SD 2.14) on a 0–10 scale, 5.31 (SD 2.23) out of 10 for sleep quality, and 10.51 (SD 3.26) out of 15 for work-related exhaustion. Stress was significantly correlated with work-related exhaustion (*r* = 0.53), and both stress and work-related exhaustion were correlated with sleep quality (inverse) and WCFS. Sleep quality was inversely correlated with workplace cognitive failure (*r* = −0.27).

Table 4 compares cognitive failure symptoms and well-being and emotional factors by practice setting. Accounting for differences in age, nurses employed in different care settings were similar in WCFS (F_4,575_ = 2.17, *p* = 0.07), sleep quality (F_4,640_ = 0.50, *p* = 0.74), stress (F_4,640_ = 1.44, *p* = 0.22), and work-related exhaustion (F_4,591_ = 2.25, *p* = 0.06). Practice setting differences were observed in secondary trauma (F_4,609_ = 3.69, *p* < 0.01). Planned comparisons with Bonferroni correction to significance testing identified nurses in Emergency/ICU settings (adjusted M = 2.40, SE = 0.10) with significantly higher ratings than those in Primary/ambulatory (adjusted M = 1.99, SE = 0.08; *p* = 0.01) and other (adjusted M = 1.92, SE = 0.10; *p* < 0.01) practice settings. 

The path model testing the hypothesis of parallel, multiple mediation of Contact and PPE on Cognitive Failure fit the data well: χ^2^ (2) = 2.35, *p* = 0.31, CFI = 1.00, RMSEA = 0.02, SRMR = 0.01; and no outliers were detected by Cook’s distance (all d = 0.00–0.87). The model accounted for a statistically significant proportion of the variance in Cognitive Failure: R^2^ = 0.25, *p* < 0.001. All mediators were significantly inter-correlated (all p’s < 0.001). Gender was only associated with differences in Exhaustion (b = 1.35, β = 0.10, *p* = 0.02) and was unrelated to all other measurements, including WCFS (all *p* ≥ 0.10). Age was associated with stress (b = −0.64, β = −0.15, *p* < 0.001), sleep (b = 0.35, β = 0.08, *p* < 0.05), secondary trauma (b = −0.15, β = −0.08, *p* < 0.05), exhaustion (b = −0.83, β = −0.13, *p* = 0.001) and contact (b = −0.38, β = −0.19, *p* < 0.001), but not WCFS or PPE (p’s ≥ 0.10). These covariates were retained in the model as statistical control variables.

Table 5 depicts results from the path analysis examining relationships between WCFS and contact with COVID patients as well as PPE supply, and the mediating effects of stress, sleep quality, secondary trauma, and work-related exhaustion. There was evidence of a relation between Contact and Cognitive Failure due to secondary effects of stress, work-related exhaustion and secondary trauma (cumulative indirect effect = 1.30, z = 6.33, *p* < 0.001). Notably, after accounting for these complex relations there was no significant direct effect of Contact (b = −0.31, β = −0.04, *p* = 0.42), and approximately 81% of the cumulative effect of Contact on Cognitive Failure was due to intervening secondary effects. All mediated effects were statistically significant, except for sleep (Table 5). Evaluating the proportional contribution of each mediator, the majority of the cumulative effect of Contact on Cognitive Failure is attributable to Secondary Trauma (44.6%), followed by Work-related Exhaustion (23.99%). Thus, H1, that Contact would be positively associated with WCFS, was not supported since direct effects were not significant. Instead, the main effects of Contact on WCFS were mediated by stress, secondary trauma, and work-related exhaustion in support of hypotheses H1a, H1c, and H1d. 

In contrast, the effects of PPE access on Cognitive Failure were distributed between direct and indirect effects (Table 5). The same set of variables significantly mediated the effect of PPE on Cognitive Failure (cumulative indirect effect = −2.10, z = −5.22, *p* < 0.001), and after accounting for this, there remained a significant direct effect: b = −1.22, β = −0.13, *p* = 0.001. This indicates that 41.97% of the effect of PPE availability on Cognitive Failure can be attributed to secondary emotional and psychological effects, but that PPE alone independently has a substantial impact that accounts for 58.03% of its cumulative effect. Of the portion that was mediated, all variables were statistically significant, except for sleep. Although the overall mediated effect size was smaller for PPE (41.97%) as compared to Contact (80.69%), a similar pattern emerged that Secondary Trauma and Work-related Exhaustion accounted for proportionally more variance than the other mediators considered. The hypothesis that PPE would be positively associated with WCFS (H2) was supported (main effect) as were hypotheses supporting the indirect effects of stress (H2a), secondary trauma (H2c), and work-related exhaustion (H2d).

## 4. Discussion

The aim of this study was to identify factors associated with workplace cognitive failure in a sample of nurses working in the early months of the COVID-19 pandemic. Using the Conservation of Resources as a theoretical framework, we hypothesized that nurses’ physical exposures, i.e., frequent contact with COVID-19 patients and availability of PPE, would each be directly related to cognitive failure. Moreover, the hypothesized relationships would be mediated by the psychosocial factors stress, sleep quality, secondary trauma, and work-related exhaustion, which would contribute to the resource loss reflected in cognitive failure. Our path model revealed that, for contact with COVID patients, essentially all its effect on cognitive failure was due to the indirect effects of all the mediators except sleep quality. For PPE, the model also revealed indirect effects on cognitive failure for all mediators except sleep quality. However, more than half the effect of PPE availability on cognitive failure was direct and independent of secondary emotional and psychological symptoms.

These findings contribute to the growing body of literature on the negative effects on the physical and psychological health of nurses providing direct patient care during the COVID-19 pandemic [10,11,12,54,55]. To the best of our knowledge, this is the first study to show the impact of caring for COVID patients on nurses’ cognitive failure at work. Cognitive failure in our sample was common, endorsed to some degree by more than 75% of nurses. A comparison of cognitive failure scores between nurses in different practice settings did not reveal significantly more cognitive failure among those working in emergency/ICU units, as expected. Emergency/ICU nurses did, however, have significantly more frequent contact with COVID-19 patients and they experienced more secondary trauma compared to colleagues in other settings. Secondary trauma was also an important mediator, accounting for most of the effect of contact with COVID patients (44.61%) on cognitive failure. While stress and work-related exhaustion did not differ across practice settings, each mediated the relationship between contact with COVID patients and cognitive failure. It is interesting that frequent contact with COVID-19 patients alone was not significantly associated with cognitive failure. In contrast to Wallace & Chen (2005) [21] and Lapierre et al. (2012) [36], this suggests that a heavy work burden alone does not lead to cognitive failure. While an overload of tasks or task complexity can lead to resource depletion and ultimate cognitive failure [36], our study suggests that, when caring for COVID patients, this occurs via the indirect effects of physical and emotional strain. It should also be noted that this study was conducted relatively early in the pandemic (May 2020), when the strain of caring for large volumes of COVID patients was complicated by the lack of knowledge about the severity and trajectory of the novel coronavirus and best methods of treatment.

As hypothesized, we found a significant inverse relationship between PPE availability and cognitive failure. Nurses who lacked adequate PPE reported significantly higher levels of cognitive failure. As with our other exposure variable, contact with COVID patients, this relationship was mediated by stress, secondary trauma, and work-related exhaustion. Inadequate PPE may also be an extra-added element of stress for nurses, as reported in several other studies [10,12,37]. This was especially true during the early stages of the pandemic, when the current study was conducted, when healthcare systems found themselves with inadequate PPE [10,12,55] and conflicting information on best practices for PPE use [56]. Lake et al. [55] recently reported that lack of PPE was associated with increased levels of moral distress in a sample of nurses. This would support the notion that the stress of caring for seriously ill, highly infectious COVID patients could contribute to a loss of resources among nurses [35]. Importantly, though, nearly 60% of PPE’s effect on cognitive failure was direct, i.e., independent of secondary emotional and physiological symptoms. 

The possible mechanisms behind these findings are not clear. In a pre-pandemic study of 2895 nurses in Iran, occupational stress and cognitive failure were associated with nurses’ self-reports of adverse events. The authors emphasized the importance of understanding the factors that influence both occupational stress and cognitive failure [24]. In the current study, we hypothesized that the extreme stress of treating COVID patients during the early stages of the pandemic would be associated with greater perceived cognitive failure among nurses, and this was largely supported. Earlier research on the impact of the COVID-19 pandemic on nurses’ health and well-being underscored the tremendous stress that nurses were experiencing early in the pandemic [12,34]. These studies identified both frequency of contact with COVID patients and inadequate PPE as significant factors in high levels of stress among frontline nurses. One study showed a clear dose–response association between PPE and depression, anxiety, and PTSD, i.e., the more inadequate the PPE supply, the greater the prevalence of mental health problems [12]. However, it is not known whether mental health problems in the current study were related to workplace cognitive failure.

Results of the current study suggest that physical exposures to the COVID-19 virus were largely responsible for nurses’ reports of cognitive failure. Greater frequency of contact with COVID patients as well as poor PPE provision, both mediated by work-related exhaustion, support this. In further support of the COR framework, the theorized resource depletion leading to cognitive failure was also influenced by well-being (stress) and emotional strain (secondary trauma) factors. Together, these factors may have reduced nurses’ resilience, making them more susceptible to cognitive failure. In pre-COVID-19 pandemic studies, high levels of work-related stress were associated with significantly higher cognitive failure scores [23,26] and were highest in nurses who reported adverse patient events [26]. A study of Korean nurses also showed a positive association between job stress and cognitive failure among hospital nurses, with cognitive failure increasing the odds of experiencing an adverse patient event [25]. A review of secondary trauma symptoms among clinicians treating trauma victims reported some studies that described “cognitive disruptions” [57] but these referred to intrusive thoughts or disruptions in one’s self-view and were not measured in terms of work-related cognitive failure. A study of nursing home workers in Spain during the COVID-19 pandemic found higher levels of secondary trauma among workers exposed to greater contact with COVID-19-patients as well as inadequate PPE [58]. That study did not focus specifically on nurses and did not measure cognitive failure.

While sleep quality might also be considered a physical/well-being factor, it did not mediate the effects of either contact with COVID patients or PPE on cognitive failure. Previous research has reported mixed results on the possible relationship between sleep/fatigue and cognitive failure among nurses. A pilot study of 28 nurses found no association between long work hours, shift work, and workplace cognitive failure [59]. However, a study of 100 nurses found that shift work, poor sleep quality, and decreased alertness while awake were directly related to impaired cognitive performance [60]. Poor sleep quality and insomnia have been reported among nurses treating COVID patients in China [4,16] and Italy [8]. However, none of those studies examined cognitive failure.

To better understand the possible mechanisms behind nurses’ workplace cognitive failure, future research should perhaps focus on measuring biomarkers associated with stress and cognition. Previous research has reported an association between levels of biomarkers of stress and inflammation in highly stressed groups of nurses [61] and emergency medicine residents [17]. Both studies revealed associations between biomarker levels and job performance, measured by adverse patient events [61] and near-misses [17]. However, neither of those studies measured self-reports of cognitive failure. A study of trauma-exposed refugees reported higher levels of brain derived neurotrophic growth factor (BDNF) and nerve growth factor (NGF), both of which were related to mental health outcomes [18]. These neurotrophins, biomarkers of neuroplasticity, could potentially affect cognitive function and ability in frontline nurses. Future studies should examine these physiological markers, focusing on practice sites where cognitive failure levels are highest.

With the advent of the Delta variant, the COVID-19 pandemic currently shows no signs of diminishing in its impact on our public health and healthcare institutions. Recent media reports describe a nursing corps “in crisis,” describing them as physically and emotionally exhausted and feeling like “forgotten soldiers” [62]. As the healthcare system risks losing many qualified nurses to the strain of the pandemic [62], the need to better understand factors that support their health and well-being has never been greater. 

### Limitations

This study has several limitations that need to be considered. It was cross-sectional in design and causality between work-related factors and cognitive failure cannot be determined. The study utilized a convenience sample that was not necessarily representative of all nurses in Michigan and results may not be generalizable to nurses in other geographic areas. However, our sample was similar to the total population of Michigan nurses in terms of gender and ethnicity. Due to the use of snowball sampling, an exact response rate could not be calculated. Workplace cognitive failure was self-reported by the nurses and our study did not include any administrative data or nurses’ self-reports documenting adverse events or near-misses. We used single-item measures for nurses’ stress and sleep quality. While these have been shown to have good validity [38,39], multi-scale items might offer more nuanced and informative data. Finally, this study examined a limited number of variables. While results shed light on factors associated with nurses’ cognitive failure, further studies examining the role of additional factors are warranted. For example, it would be interesting to explore the possible role of social support from colleagues and family members, known to enhance resilience to job strain [63] as a protective factor against cognitive failure. Social support has also been associated with lower rates of turnover among nurses during the COVID-19 pandemic [64].

## 5. Conclusions

To the best of our knowledge, this study is the first to identify factors associated with cognitive failure in nurses working with the care of COVID-19 patients. Workplace cognitive failure in nurses during the early months of the current pandemic was associated with the physical and psychological strain of caring for COVID patients. This increased strain, in combination with the new and unknown nature of the pandemic, may have been a potential breeding ground for errors or near-misses in patient care. To reduce cognitive failure and safeguard nurses’ ability to work effectively, healthcare organizations should be diligent in their efforts to provide nurses with the necessary support and personal protective equipment to enable them to maintain their well-being and to work safely. To reduce the risk of cognitive failure, healthcare organizations need to provide nurses with protective equipment and work environments that allow nurses to strengthen their resilience to physically and psychologically taxing work conditions. Regular monitoring of nurses’ work stress and interventions to enhance their emotional well-being are warranted. 

## Figures and Tables

**Figure 1 ijerph-18-10394-f001:**
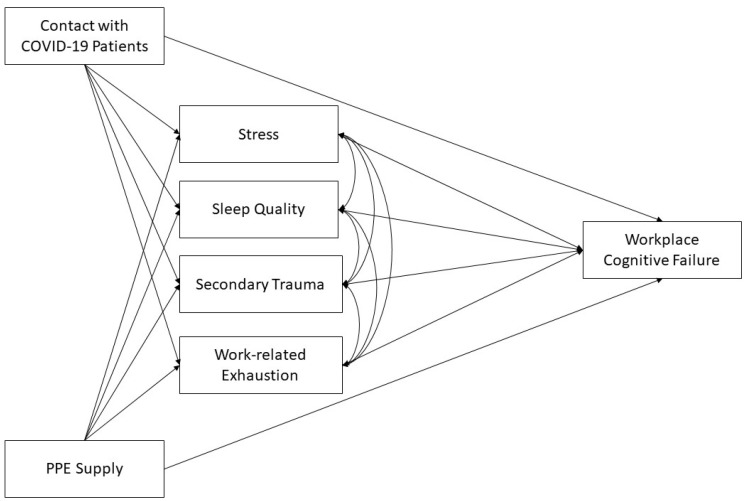
Conceptual model of hypothesized relationships between physical exposures and workplace cognitive failure and the mediating effects of psychosocial factors.

**Table 1 ijerph-18-10394-t001:** Characteristics of study participants (n = 695).

	N ^a^ (%)
Age (years)	
18–24	22 (3.2)
25–34	123 (17.8)
35–44	167 (24.1)
45–54	153 (22.1)
55–64	176 (25.4)
65–74	42 (6.1)
75 and above	5 (0.7)
Gender	
Males	44 (6.4)
Females	644 (93.6)
Race	
White	611 (87.9)
Black/African American	34 (4.9)
Other	50 (7.2)
Practice setting	
Acute care/Inpatient	220 (32.8)
Emergency/Intensive care	102 (15.2)
Pediatrics/Obstetrics	60 (8.9)
Primary/Ambulatory care	185 (27.6)
Hospice/Home care/Long-term care	32 (4.8)
Mental health	18 (2.7)
Academic	54 (8.0)
Contact with COVID-19 patients	
Never	127 (19.0)
Seldom	273 (40.8)
Often	142 (21.2)
Very Often	127 (19.0)
Adequate PPE provided by workplace	
Not at all	49 (7.5)
Not really	114 (17.4)
Somewhat	238 (36.3)
Definitely	200 (30.5)
Not applicable	55 (8.3)
Secondary trauma from patients cared for	
Never	204 (33.0)
Rarely	201 (32.5)
Sometimes	172 (27.8)
Often	33 (5.3)
Very often	9 (1.5)

Note: COVID-19, Coronavirus disease; PPE, Personal protective equipment; ^a^ Numbers do not add to group totals due to missing values.

**Table 2 ijerph-18-10394-t002:** Discrete study variables by practice setting (n = 695).

	Practice Setting					
	Acute Care/Inpatient	Emergency/ICU	Pediatrics/OB	Primary/Ambulatory	Other	Overall *p*-Value
	N (%)	N (%)	N (%)	N (%)	N (%)	
Age						<0.001
<45 years	120 (55.0)	66 (65.3)	22 (36.7)	61 (33.2)	32 (30.8)	
≥45 years	98 (45.0)	35 (34.7)	38 (63.3)	123 (66.8)	72 (69.2)	
Gender						0.07
Males	19 (8.7)	8 (7.8)	0 (0.0)	9 (4.9)	4 (3.9)	
Females	199 (91.3)	94 (92.2)	60 (100.0)	176 (95.1)	99 (96.1)	
Contact with COVID-19 patients						<0.001
Never	34 (15.5)	3 (3.0)	15 (25.0)	33 (17.8)	41 (39.8)	
Seldom	87 (39.7)	16 (16.0)	35 (58.3)	97 (52.4)	37 (35.9)	
Often	52 (23.7)	25 (25.0)	7 (11.7)	40 (21.6)	18 (17.5)	
Very often	46 (21.0)	56 (56.0)	3 (5.0)	15 (8.1)	7 (6.8)	
Workplace provided adequate PPE						0.17
No, not at all	11 (5.5)	7 (7.1)	4 (7.7)	19 (11.1)	8 (10.1)	
Not really	39 (19.4)	22 (22.4)	16 (30.8)	23 (13.5)	14 (17.7)	
Somewhat	86 (42.8)	36 (36.7)	20 (38.5)	71 (41.5)	25 (31.6)	
Definitely	65 (32.3)	33 (33.7)	12 (23.1)	58 (33.9)	32 (40.5)	
Secondary trauma from patients cared for						<0.001
Never	52 (25.6)	24 (25.0)	18 (31.6)	71 (41.8)	39 (42.4)	
Rarely	75 (36.9)	23 (24.0)	17 (29.8)	51 (30.0)	34 (37.0)	
Sometimes	69 (34.0)	35 (36.5)	18 (31.6)	37 (21.8)	13 (14.1)	
Often	6 (3.0)	12 (12.5)	4 (7.0)	8 (4.7)	3 (3.3)	
Very often	1 (0.5)	2 (2.1)	0 (0.0)	3 (1.8)	3 (3.3)	

Note: COVID-19, Coronavirus disease; PPE, Personal protective equipment; ICU, Intensive care unit; OB, Obstetrics; Other practice setting includes Hospice/Home care/Long-term care, Mental health, and Academic.

**Table 3 ijerph-18-10394-t003:** Bivariate correlations between stress, work-related exhaustion, sleep quality, and cognitive failure symptoms (n = 695).

	Mean (SD)	1	2	3
1 Stress (0–10)	5.95 (2.14)	-		
2 Work-related exhaustion (3–15)	10.51 (3.26)	0.53	-	
3 Sleep quality (0–10)	5.31 (2.23)	−0.48	−0.41	
4 WCFS (15–75)	28.75 (8.66)	0.33	0.38	−0.27

Note: Variable means and standard deviations are reported for scale scores. Pearson correlations are reported; all correlations are significant at *p* < 0.001; WCFS, Workplace cognitive failure scale; higher values on WCFS indicate more symptoms of cognitive failure.

**Table 4 ijerph-18-10394-t004:** Cognitive failure symptoms, well-being, and emotional factors by practice setting (n = 671).

			Planned Group Comparisons by Practice Setting				
	Main Effect of Practice Setting		Emergency/ICU (n = 102)	Acute Care/Inpatient (n = 220)	Pediatrics/OB (n = 60)	Primary/Amb (n = 185)	Other (n = 104)
Outcome (Range)	F Test (η_p_^2^)	*p*-Value	Adjusted Mean (SE)	Adjusted Mean (SE)	Adjusted Mean (SE)	Adjusted Mean (SE)	Adjusted Mean (SE)
Cognitive failure							
WCFS sum (15–75)	2.17 (0.02)	0.07	30.63 (0.90)	29.12 (0.62)	29.52 (1.21)	27.65 (0.69)	27.64 (0.96)
Sleep quality (0–10)	0.50 (0.00)	0.74	5.10 (0.23)	5.29 (0.15)	5.06 (0.29)	5.34 (0.17)	5.47 (0.23)
Stress (0–10)	1.44 (0.01)	0.22	6.43 (0.22)	5.86 (0.14)	5.78 (0.28)	5.94 (0.16)	6.07 (0.21)
Secondary trauma (1–5)	**3.69 (0.02)**	**0.01**	**2.40 (0.10)**	2.14 (0.07)	2.18 (0.13)	**1.99 (0.08)**	**1.92 (0.10)**
Work-related exhaustion (3–15)	2.25 (0.02)	0.06	11.40 (0.33)	10.60 (0.23)	10.42 (0.43)	10.32 (0.25)	10.08 (0.35)

Note: SE, Standard error; ICU, Intensive care unit; OB, Obstetrics; Amb, Ambulatory; WCFS, Workplace cognitive failure scale; Other practice setting includes Hospice/Home-care/Long-term care, Mental health, and Academic. Higher values on WCFS and each subscale indicate more symptoms of cognitive failure. Bold entries indicate statistically significant effects of the F-test, and planned comparisons to emergency/ICU with Bonferroni correction. F-tests are reported with standardized effect size (η_p_^2^). Adjusted means, controlling for age as a covariate, are reported for each group and planned comparisons are only interpreted following a significant F-test.

**Table 5 ijerph-18-10394-t005:** Path model testing the hypothesis of parallel, multiple mediation of Contact with COVID-19 patients and PPE Availability on Workplace Cognitive Failure.

	Contact with COVID-19 Patients		PPE Availability	
	Estimate (LL, UL)	% Cumulative	Estimate (LL, UL)	% Cumulative
Total Effect	**0.99** (0.24, 1.73)		**−2.10** (−2.89, −1.31)	
Direct Effect	−0.31 (−1.06, 0.44)	19.31%	**−1.22** (−1.93, −0.51)	58.03%
Indirect Effects	**1.30** (0.89, 1.70)	80.69%	**−0.88** (−1.28, −0.48)	41.97%
Stress	**0.15** (0.02, 0.27)	9.10%	**−0.16** (−0.30, −0.01)	7.43%
Sleep	0.05 (−0.07, 0.17)	2.99%	−0.08 (−0.29, 0.12)	4.00%
Secondary Trauma	**0.72** (0.40, 1.03)	44.61%	**−0.32** (−0.53, −0.11)	15.29%
Work-related Exhaustion	**0.39** (0.17, 0.60)	23.99%	**−0.32** (−0.52, −0.12)	15.20%

Note: LL, lower limit; UL, upper limit; PPE, personal protective equipment. Bold entries are statistically significant.

## Data Availability

The data presented in this study are available on request from the corresponding author.

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
