# Peer review of "Workplace Cognitive Failure among Nurses during the COVID-19 Pandemic"

_ijerph, 2021, doi:10.3390/ijerph181910394_

Round 1
Reviewer 1 Report
Overall, this is an interesting study and quite a well-written manuscript that has the potential to shed more light on the complex relationship between different psychological factors and cognitive failure among nurses during the COVID-19 pandemics. In my opinion, the study is well designed. The aims of the study are original. The group sample size is honourable. Generally, the introduction and discussion are suitable and wide and new literature was referenced in the appropriate context.
Some points need to be considered:
Methods:
- There is no information about the bioethics commission approval number. This information should be added to the manuscript.
Discussion:
- Single questions were used to assess stress and sleep quality. There are other adequate tools that were not used in this study. In my opinion, a lack of these tools is an important limitation of this study and authors should add this information to the limitations section.
Author Response
Please see the attachment

This manuscript is a resubmission of an earlier submission. The following is a list of the peer review reports and author responses from that submission.
Round 1
Reviewer 1 Report
Overall, this is an interesting study and quite a well-written manuscript that has the potential to shed more light on the relationship between different factors (psychological and workplace) and cognitive failure among nurses during the 2 COVID-19 pandemics. In my opinion, the study is well designed. The aims of the study are original. The group sample size is honorable. Generally, the introduction and discussion are not suitable but wide and new literature was referenced in the appropriate context.
Some points need to be considered:
Title:
- I suggest modifying the title because the term "cognitive failure" is not the same as "work performance". In my opinion, the authors studied "cognitive failure" among nurses.
Introduction:
- There is a lack of a good theoretical model about the relationship between variables. In my opinion, the authors should search the literature and find the model of the relationship between different psychological factors (e.g., stress, work-related physical and emotional strain) and cognitive failure.
- There is a lack of adequate reason for the research hypotheses.
- Hypotheses should be modified: (a) first hypothesis is a non-directional hypothesis as well as a directional hypothesis (the authors should decide, with which is correct, because it is determining the further steps of statistical analysis, e.g., for non-direct hypothesis = ANOVA with post hoc or for direct hypothesis = ANOVA with planned contrasts); (b) second and third hypotheses are not good; in my opinion, the authors should verify the hypothesis of a relationship between different factors and cognitive failure in nurses (moreover, it will be very interesting to check a complex relationship between variables using a mediation model e.g., with using the PROCESS macro for SPSS by Hayes).
Materials and Methods:
- Is the authors had got the consent of the bioethics committee for the research? It is not clear. If not, there is a huge problem to publish a study without that consent.
- In my opinion, there is a lack of good division of variables; in this study, the main independent variable is practice setting; other variables (e.g., stress, sleep quality, etc.) are additional independent variables; demographic factors are control variables.
- Single questions were used to assess stress and sleep quality. There are adequate tools such as the Pittsburgh Sleep Quality Index (PSQI) or the Epworth Sleepiness Scale (ESS), or the Short Stress State Questionnaire (SSSQ). In my opinion, a lack of these tools is an important limitation of this study.
- Was sample size calculated using G-Power or another statistical tool?
Results:
- Age is a continuous variable and the authors should conduct ANOVA with post hoc to check the differences between groups in this factor.
- In the Tables presenting results of ANOVA should be reported value of statistical tests and effect size (sometimes a statistical power), not only p-value.
- If there were differences in age between groups, the authors should use ANCOVA with co-varying age to check the differences between groups in cognitive failure.
Discussion:
- Discussion should be corrected after conduct a new statistical analysis.
- The authors should address a good psychological theoretical model of the relationship between variables and explain their results in an appropriate context.
General, in my opinion, this manuscript should be modified:
- I propose to consider a complex theoretical and statistical model (using mediations model e.g., with psychological factors as mediators between frequent contact with COVID and inadequate PPE, and cognitive failure in nurses.
- The article in this form with applied statistical analysis is very poor in new knowledge.
Reviewer 2 Report
Dear authors,
Thank you so much for submitting this interesting investigation. The information and its presentation are quite good, even though I would like to give you some commentaries that I hope you find useful:
- Information about the scales used is provided, but there is no data about the Quality Work Competence questionnaire's reliability.
- In tables is a little bit difficult to read the info, maybe an alignment to the left for some lines (e.g. age ranges) would make it easier. Also, please check if the word "Note" is needed to be included at the table feet.
- Some participants were unemployed or retired. Were those data taking into account in the investigation? From my point of my view, they should be retired from the data (if they were not working while pandemic has been present).
- Was the division at 45 years (L208, table 2) arbitrary?
- L222: retire the last comma
- Page 10 and followings: please check the page numeration
- Following steps or investigation lines would be interesting to be added. For example, could the social support affect the results?
Thank you in advance,
Best regards